# Dimerization and dynamics of human angiotensin-I converting enzyme revealed by cryo-EM and MD simulations

Jordan M Mancl[1], Xiaoyang Wu[1], Minglei Zhao[2], Wei-Jen Tang[1]*

[1]Ben-May Department for Cancer Research, University of Chicago, Chicago, Illinois, United States; [2]Department of Biochemistry and Molecular Biology, University of Chicago, Chicago, Illinois, United States

## eLife Assessment

This study shows, for the first time, the structure and snapshots of the dynamics of the full-length soluble Angiotensin-I converting enzyme dimer. The combination of structural and computational analyses provides **compelling** evidence that reveals the conformational dynamics of the complex and key regions mediating the conformational change. This **fundamental** work illustrates how conformational heterogeneity can be used to gain insights into protein function.

*For correspondence:
wtang@bsd.uchicago.edu

Competing interest: The authors declare that no competing interests exist.

**Abstract** Angiotensin-I converting enzyme (ACE) regulates the levels of disparate bioactive peptides, notably converting angiotensin-I to angiotensin-II and degrading amyloid beta. ACE is a heavily glycosylated dimer, containing four analogous catalytic sites, and exists in membrane-bound and soluble (sACE) forms. ACE inhibition is a frontline, FDA-approved, therapy for cardiovascular diseases yet is associated with significant side effects, including higher rates of lung cancer. To date, structural studies have been confined to individual domains or partially denatured cryo-EM structures. Here, we report the cryo-EM structure of the glycosylated full human sACE dimer. We resolved four structural states at 2.99 – 3.65 Å resolution which are primarily differentiated by varying degrees of solvent accessibility to the active sites and reveal the full dimerization interface. We also employed all-atom molecular dynamics (MD) simulations and heterogeneity analysis in cryo-SPARC, cryoDRGN, and RECOVAR to elucidate the conformational dynamics of sACE and identify key regions mediating conformational change. We identify differences in the mechanisms governing the conformational dynamics of individual domains that have implications for the design of domain-specific sACE modulators.

## Introduction

Angiotensin-I converting enzyme (ACE) is an M2 clan zinc metalloprotease (EC3.4.15.1) located on the plasma membrane and in the extracellular milieu (*Bernstein et al., 2013*; *Lubbe et al., 2020*; *Turner and Hooper, 2002*). ACE plays a crucial role in the renin-angiotensin system by converting angiotensin I to the potent vasoconstrictor angiotensin II, and in the kinin-kallikrein system by degrading bradykinin, a potent vasodilator. Inhibiting ACE is a primary approach for treating hypertension and is a proven therapy for heart failure, diabetic nephropathy, and other cardiovascular and renal dysfunctions (*Herman et al., 2024*). However, ACE inhibition has known side effects, e.g., dry cough and angioedema, and has been linked to higher rates of lung cancer (*Wu et al., 2023*; *Yao et al., 2023*). This is partly because ACE also degrades a diverse range of peptides, resulting in complex physiological outcomes due to the interplay in altering levels of these peptides (*Herman et al., 2024*; *Semis*

*et al., 2019*; *Zheng et al., 2022*; *Bernstein et al., 2018*; *Danziger et al., 2023*). Furthermore, ACE degrades amyloid β (Aβ), which is associated with the progression of Alzheimer's diseases (*Hampel et al., 2021*; *Hemming and Selkoe, 2005*; *Zou et al., 2009*; *Zhang et al., 2023*). Overexpression of ACE in myelomonocytes substantially reduced Aβ load and prevented Alzheimer's disease-like cognitive decline in mice (*Bernstein et al., 2014*). However, the brains of some Alzheimer's disease patients showed upregulation of the renin-angiotensin system, which led to deleterious effects, and ACE inhibition was beneficial (*Gouveia et al., 2022*; *Wright et al., 2013*; *Qiu et al., 2013*). These studies indicate that global ACE inhibition could be either beneficial or harmful for individual Alzheimer's patients and underscore a need for better understanding the molecular mechanisms governing ACE function and substrate selectivity to develop more selective, improved ACE inhibitors and expand ACE-based therapies.

ACE is a 180 kDa heavily N-linked glycosylated type 1 transmembrane protein with two homologous catalytic domains residing at the N- and C-terminal ends (ACE-N and ACE-C) in its extracellular region (*Bernstein et al., 2013*; *Turner and Hooper, 2002*; *Riordan, 2003*). The ACE extracellular region can be released via proteolytic cleavage near the transmembrane region by ADAM (*a d*isintegrin *a*nd *m*etalloproteinase) family protease(s), generating soluble ACE (sACE) and ACE dimerization can regulate this process (*English et al., 2012*; *Hooper and Turner, 2002*; *Kost et al., 2003*; *Webers et al., 2024*). ACE belongs to the cowrin (cowry-like) family of zinc metallocarboxypeptidases because its catalytic domain is ellipsoidal and contains a long, deep, yet narrow catalytic cleft (*Gomis-Rüth, 2008*). ACE is known to cleave short peptides, such as angiotensin I (decapeptide) and bradykinin (nonapeptide) by removing the C-terminal two amino acid residues. Different from the funnelin family of metallocarboxypeptides that contains a funnel-like cavity, the catalytic cleft of ACE can accommodate entire oligopeptides and ACE has been shown to cleave multiple discrete sites on Aβ outside the C-terminal end (*Zou et al., 2009*). Thus, it is more accurate to refer to ACE as an endopeptidase with dipeptidyl carboxylase activity.

Extensive crystallographic studies of ACE-N or ACE-C alone and in complex with various inhibitors and a catalytic product, angiotensin-II, have elucidated the overall structures of ACE catalytic domains and molecular details of the catalytic pockets (*Lubbe et al., 2020*; *Acharya et al., 2024*; *Corradi et al., 2006*; *Masuyer et al., 2012*; *Natesh et al., 2003*; *Acharya et al., 2003*). The structures indicate that each domain is comprised of three sub-domains and offer insight for the substrate preferences around cleavage sites and the promiscuity of each domain toward short peptide substrates. Furthermore, they reveal the orientation of substrate binding sites around the scissile bond, indicating that the entire oligopeptide substrates need to be engulfed into the catalytic chamber of ACE catalytic domain (*Acharya et al., 2024*; *Masuyer et al., 2012*; *Natesh et al., 2003*).

However, the catalytic pocket of ACE in the closed state is not large enough to accommodate the entrance of all its known substrates, particularly Aβ, to permit the observed cleavage pattern. Thus, similar to other Aβ-degrading proteases, such as insulin degrading enzyme (IDE), presequence protease (PreP), and neprilysin (NEP), ACE must undergo a large-scale open-closed transition to capture and degrade larger peptide substrates such as Aβ (*Liang et al., 2022*; *Malito et al., 2008*; *Zhang et al., 2018*). Recent open state structures of sACE-N, sACE monomer, and a sACE-N dimer, along with molecular dynamics (MD) simulations of sACE-C, have begun to reveal ACE conformational dynamics, though they remain under-studied (*Cozier et al., 2021*; *Lubbe et al., 2022*; *Phan et al., 2020*).

The extracellular domains of ACE are readily dimerized in the membrane-bound and soluble forms. Both the ACE-N and ACE-C domains within the ACE monomer are catalytically active and each has its own substrate preferences (*Semis et al., 2019*; *Zou et al., 2009*; *Bernstein et al., 2011*). For example, ACE-N can convert Aβ1–42 to Aβ1–40, which is less amyloid fibril-prone than Aβ1–42 but ACE-C cannot (*Zou et al., 2009*). Accumulating evidence supports a complicated interplay between ACE-N and ACE-C within the ACE dimer. Enzyme kinetics analysis suggests negative cooperativity between two catalytic domains (*Binevski et al., 2003*; *Rice et al., 2004*; *Skirgello et al., 2005*). However, ACE also exhibits positive synergy toward Aβ cleavage and allostery to enhance the activity of its binding partner, bradykinin receptor (*Zou et al., 2009*; *Erdös et al., 2010*). Recently, cryo-EM structures of soluble dimeric apo-sACE were reported *Lubbe et al., 2022*; however, due to the poor resolution/denaturation of sACE-C, only full-length sACE monomer or dimeric sACE-N was reported yet the conformations preclude full-length sACE dimerization due to the severe steric crash between

sACE-C domains. Here, we present four discrete cryo-EM structures of full-length human sACE that contain all four catalytic domains (*Figure 1A*). This is followed by cryo-EM heterogeneity analysis and all-atom MD simulations. Together, our analyses provide structural insights into domain interactions, open-closed transitions of the catalytic domain, and conformational dynamics of soluble dimeric ACE.

## Results

### Structures of the full-length sACE dimer determined by single particle cryo-EM

Soluble human ACE (sACE), which includes the entire extracellular region of ACE, was expressed in HEK293F cells and purified from culture supernatant using an ion exchanger (Source Q) followed by size exclusion chromatography (Superdex S200). Buffer conditions suitable for single particle cryo-EM analysis were identified and optimized using differential scanning fluorometry (*Figure 1—figure supplement 1*, *Supplementary file 1*). Purified sACE was then vitrified using a Vitrobot, and a dataset of ~3600 micrographs was collected on a 300 keV Titan Krios equipped with a Gatan K3 camera at the University of Chicago Advanced Electron Microscopy Facility (*Supplementary file 2*). The data was processed in cryoSPARC v4.4.0 (*Figure 1—figure supplement 2*). After filtering ~1.1 million picked particles to ~378 K using 2D and 3D classification, three classes were identified (*Figure 1*, *Figure 1—figure supplement 2*). Of these, ~268 K particles mapped to a dominant 2-domain class reconstructed as an sACE-N dimer, while ~21 K particles mapped to a minor 3-domain class containing two sACE-N and only one sACE-C, reminiscent of the recently published sACE cryo-EM structure (*Lubbe et al., 2022*). The remaining 89 K particles mapped to a full-length sACE dimer containing two sACE-N and two sACE-C, which was refined to 3.65 Å resolution (*Figure 1C*, *Figure 1—figure supplements 2 and 3*, *Figure 1—video 1*, *Supplementary file 2*).

The partial denaturation of macromolecules during vitrification has been attributed mainly to repeated exposure of the macromolecules to the air-water interface (*Glaeser, 2018*; *Noble et al., 2018*). The sACE-C domain was shown to be less stable than the sACE-N domain (*Voronov et al., 2002*), so we hypothesized that repeated exposure to the air-water interface could cause preferential denaturation of sACE-C. We have previously shown that preferential denaturation of human PreP occurred in one of its two homologous domains and that faster vitrification using Chameleon significantly prevented such preferential denaturation (*Liang et al., 2022*). Therefore, we collaborated with the National Center for CryoEM Access and Training (NCCAT) to prepare sACE grids using Chameleon (*Darrow et al., 2019*). Chameleon preparation reduced the time the sample spends on the grid before freezing by more than 10-fold. We collected a new dataset of ~18,000 micrographs, and our 2D classification revealed a higher percentage of particles for four-domain classes than those prepared using the Vitrobot (*Figure 1B*). However, our 3D classification still showed the existence of structures of the sACE-N only dimer and those containing two sACE-N and one sACE-C (*Figure 1—figure supplement 4*). This indicates that the reduction in vitrification time using Chameleon reduced but did not prevent preferential denaturation.

From these data, we obtained a map for a consensus ACE structure that achieved a resolution of 2.8 Å. However, while the core region showed excellent map quality, the map quality for the dynamic regions corresponding to the open-close transition of ACE catalytic domains was poor, suggesting multiple discrete conformational states exist. We thus performed 3D classification and reconstructed three distinct conformational states of full-length apo-ACE at resolutions of 2.99 Å, 3.05 Å, and 3.15 Å (*Figure 1C*, *Figure 1—figure supplements 4 and 5*, *Figure 1—videos 2–4*, *Supplementary file 2*). We refer to these ACE dimer structures by their resolutions, namely sACE-2.99, sACE-3.05, and sACE-3.15, along with the structure from data obtained using the Vitrobot, sACE-3.65. Given the rich structural information from more than 50 reported crystal structures of ACE and one recently reported cryo-EM study of ACE, below we focus on the new insights that our sACE dimer structures offer (*Lubbe et al., 2020*; *Acharya et al., 2024*; *Masuyer et al., 2012*; *Lubbe et al., 2022*; *Corradi et al., 2006*; *Natesh et al., 2003*).

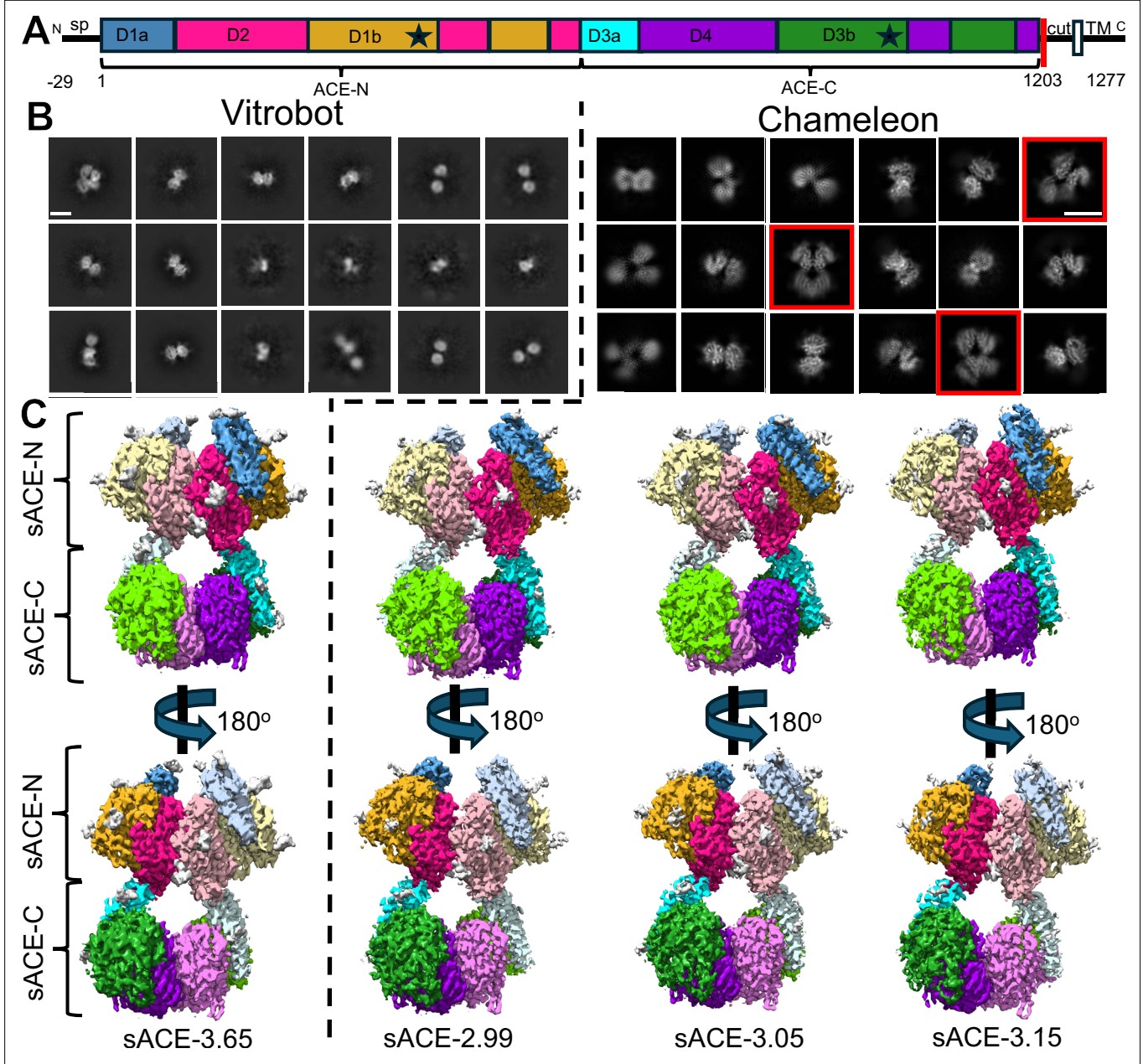

**Figure 1.** Cryo-EM analysis of human sACE dimer. (**A**) Diagram for the key features of domain on primary sequence of human ACE. The construct used in this study contains the first 1235 residues, which we refer to as the soluble region of ACE (sACE); however, the first 29 residues comprise a signal. By convention, sACE is labeled based on the mature, processed peptide (*Soubrier et al., 1988*). D1a and D3a domains, also called 'lid', encompass residues 1–98 and aa 616–696, respectively. D1b and D3b domains each have two discrete segments; residues 263–436 and 496–574 for D1b and residues 868–1031 and 1091–1171 for D3b. D2 encompasses three discrete segments, residues 99–262, 437–495, and 575–615 while D4 domain encompasses residues 697–867, 1032–1090, and 1172–1202. Sp = signal peptide. Asterisk = zinc binding motif. Colored by sub-domain: D1a blue, D1b gold, D2 magenta, D3a cyan, D3b green, D4 purple. (**B**) 2D classification of human sACE particles from grids made by vitrobot (360 pixels box size) and Chameleon (256 pixels box size). Clear four domain classes visible in the Chameleon-derived classification are boxed in red, similar views are lacking in the vitrobot dataset. White scale bar measures 100 Å. (**C**) Full-length sACE 3D volumes, colored by sub-domain as in (**A**). Chain B sub-domains are depicted as lighter tones of their chain A counterparts. Glycan density is shown in gray. See *Figure 1—figure supplement 2* for vitrobot-prepared data processing details, *Figure 1—figure supplement 4* for Chameleon-prepared data processing details and *Supplementary file 2* for data refinement statistics.

The online version of this article includes the following video and figure supplement(s) for figure 1:

**Figure supplement 1.** DSF optimization of sACE grid-making conditions.

*Figure 1 continued on next page*

*Figure 1 continued*

**Figure supplement 2.** Processing workflow for dataset using Vitrobot-prepared grids.

**Figure supplement 3.** Assessment of map quality from Vitrobot prepared grids.

**Figure supplement 4.** Processing workflow for dataset using Chameleon-prepared grids.

**Figure supplement 5.** Assessment of map quality from Chameleon prepared grids.

**Figure 1—video 1.** Overview of sACE-3.65 structure.

https://elifesciences.org/articles/106044/figures#fig1video1

**Figure 1—video 2.** Overview of sACE-2.99 structure.

https://elifesciences.org/articles/106044/figures#fig1video2

**Figure 1—video 3.** Overview of sACE-3.05 structure.

https://elifesciences.org/articles/106044/figures#fig1video3

**Figure 1—video 4.** Overview of sACE-3.15 structure.

https://elifesciences.org/articles/106044/figures#fig1video4

## The overall structures of sACE dimers and structural insight into dimerization

Soluble ACE has two homologous catalytic domains: sACE-N (amino acids 1–615) and sACE-C (amino acids 616–1202). Based on the comparison of various catalytic domains of our ACE dimer structures, we subdivide each catalytic domain of sACE into three subdomains: D1a, D1b, and D2 for sACE-N, and D3a, D3b, and D4 for sACE-C (*Figure 1A*). These three subdomains in either sACE-N or sACE-C form a catalytic chamber. The catalytic zinc ion is coordinated by the canonical HEXXH motif residing in the D1b and D3b subdomains (*Figure 1A*). Four sACE dimer structures manifest as pseudo-symmetric C2 homodimers across the dimer interface. The dominant symmetry-breaking features are the degrees of displacement between D1 and D2 or between D3 and D4, leading to changes in solvent accessibility to the catalytic pockets.

Based on the openness of the catalytic chamber, defined as the distance between the D1/3 a and D2/4 helices that gate access, we classified sACE-N into three conformational states: open (O), intermediate open (I), and closed (C) (*Figure 2A*); sACE-C is significantly less open than sACE-N in our structures and is only classified into intermediate (I) and closed (C) states (*Figure 2B*). The four sACE dimer structures differ mostly by the degrees of openness of their four catalytic domains (*Figure 2C and D*). Our sACE-2.99 and sACE-3.65 structures are globally similar, both contain sACE-N domains in the open and intermediate states and sACE-C domains in the closed and intermediate states. In sACE-3.05, both sACE-N domains adopt an open state, while the sACE-C domains adopt an intermediate and closed state. In the sACE-3.15 structure, one subunit has both sACE-N and sACE-C adopting a closed conformation, while the other subunit has an open sACE-N domain and an intermediate sACE-C domain.

Our four cryo-EM structures of sACE provide the structural basis for sACE dimerization. sACE dimerization is mediated by interfaces between D2 subdomains and D4 subdomains of sACE monomers. The D2 and D4 subdomains generally have much lower B-factors and higher local resolution estimates than the D1 and D3 subdomains (*Figure 1—figure supplement 3*, *Figure 1—figure supplement 5*, *Figure 2—figure supplement 1*). The interaction of the D4-D4 subdomains is analogous to the D2-D2 interaction, yielding an RMSD of ~1 Å² when aligned (*Figure 3A*). The primary source of variation is a general rigid body shift away from the interaction interface for non-interacting regions in the D4-D4 region compared to the D2-D2 region. The observed D2-D2 interface is consistent with one of three possible contacts between sACE-N in the non-crystallographic symmetry mates from previous crystallographic analysis of sACE-N and the previously published cryo-EM structure (*Figure 3—figure supplement 1*). In total, the interface between sACE monomers buries a surface area of approximately 1900 Å². Although sACE-N and sACE-C are structurally analogous, their respective interfaces (sACE-N/N and sACE-C/C) are not equivalent. The sACE-N/N interface is larger and stronger, burying a surface area of approximately 1,150 Å², compared to a buried surface area of approximately 750 Å² for the sACE-C/C interface.

The interface between sACE-N is formed by a hydrophobic core, anchored by a Y465 π-stacking interaction, and flanked by salt bridges formed by R453 and R458 as well as E212 and E219 along

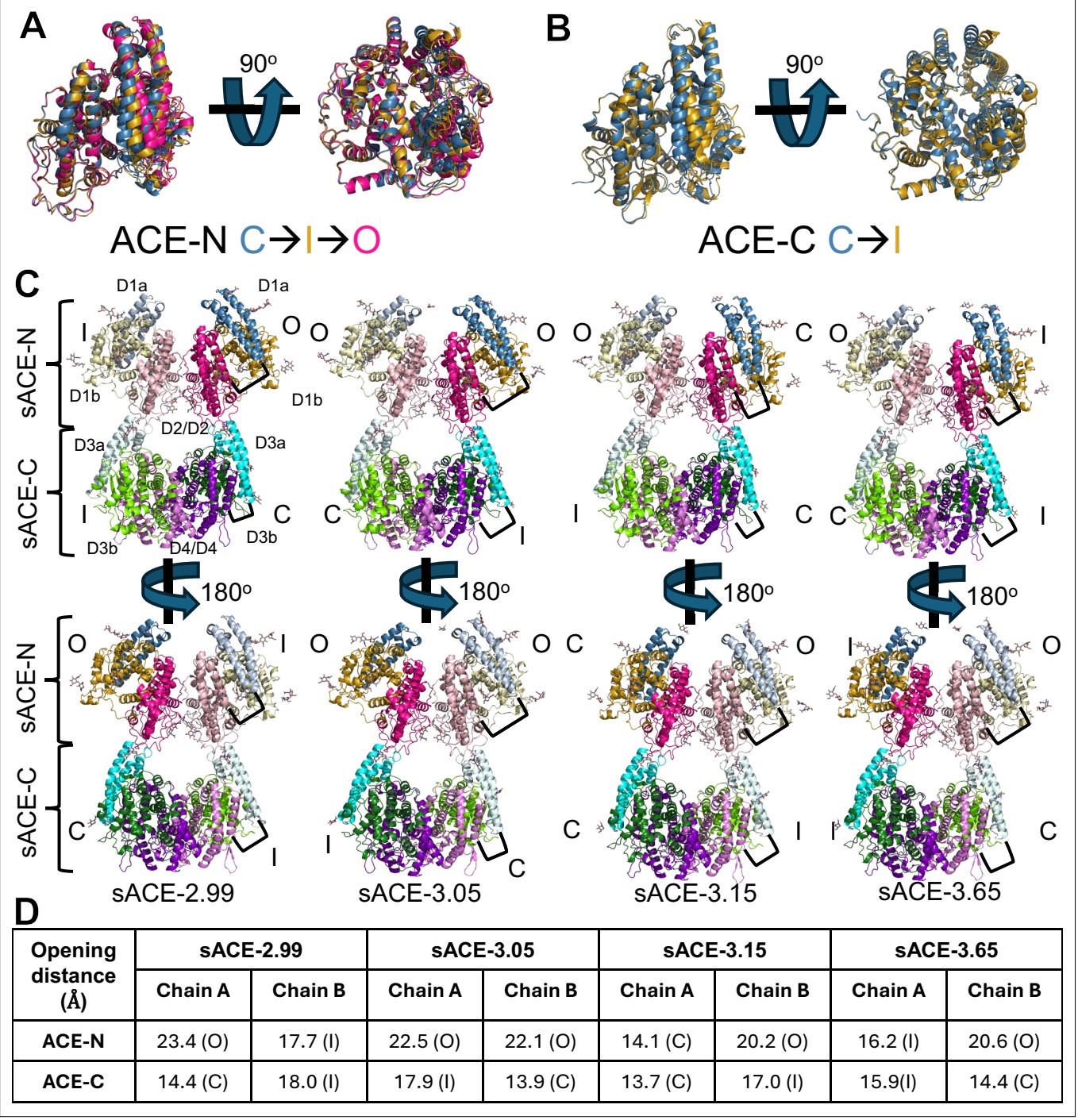

| Opening distance (Å) | sACE-2.99 | | sACE-3.05 | | sACE-3.15 | | sACE-3.65 | |
|---|---|---|---|---|---|---|---|---|
| | Chain A | Chain B | Chain A | Chain B | Chain A | Chain B | Chain A | Chain B |
| ACE-N | 23.4 (O) | 17.7 (I) | 22.5 (O) | 22.1 (O) | 14.1 (C) | 20.2 (O) | 16.2 (I) | 20.6 (O) |
| ACE-C | 14.4 (C) | 18.0 (I) | 17.9 (I) | 13.9 (C) | 13.7 (C) | 17.0 (I) | 15.9(I) | 14.4 (C) |

**Figure 2.** Overall structure of human sACE. (**A**) Overlay comparing sACE-N states highlighting the structure differences between the closed 'C', intermediate 'I', and open 'O' states. (**B**) Overlay comparing sACE-C states highlighting the structure differences between the closed 'C' and intermediate 'I' states. We define the state based on the distance between the edge of the D2/4 domain bordering the catalytic cleft (residues 121–126 or 721–726) and the tip of the D1/3 a region (residues 41–51 or 647–657): closed <15 Å, intermediate >15 Å and <19 Å, open >19 Å. (**C**) Overall dimer comparisons. Black bracket depicts the D1/3a-D2/4 distance measurement used to define the state of each domain, as described above. (**D**) Table of openness measurements for each domain per structure.

The online version of this article includes the following figure supplement(s) for figure 2:

**Figure supplement 1.** B factor comparison of all structures.

**Figure supplement 2.** sACE-N/N dimers compared to previously published structures.

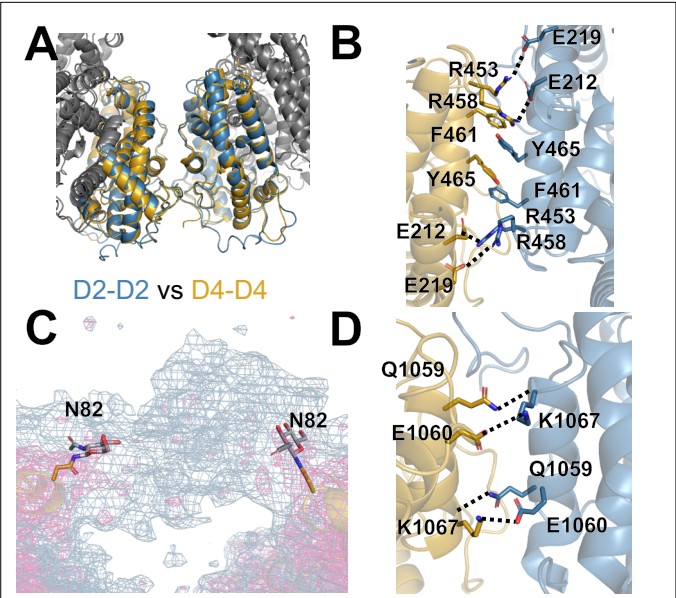

**Figure 3.** sACE dimerization interfaces. (**A**) Overlay comparing sACE-N/N interface (blue) and sACE-C/C (gold) interfaces. Interfaces adopt the same secondary structure but interacting residues vary between them. (**B**) Residue-specific interactions at the sACE-N/N interface, see text for details. (**C**) Unsharpened Coulomb potential density map (blue) showing density corresponding to glycan-glycan interaction from N82 as part of sACE-N/N interface. A sharpened map is shown in magenta for reference. (**D**) Residue-specific interactions in the sACE-C/C interface, see text for details.

The online version of this article includes the following figure supplement(s) for figure 3:

**Figure supplement 1.** sACE exhibits a continuous gradient of structural heterogeneity.

with a collection of polar and van der Waals contacts, which is consistent with the previous mutational studies of Y465 (*Figure 3B*; *Danilov et al., 2011*). The dimer interface between two sACE-N domains is completed by glycan-glycan interactions between the glycans attached to N82, which is consistent with the role of glycosylation in sACE dimerization (*Figure 3C*; *Kost et al., 2000*). However, the central residues that form the hydrophobic core of the interface between sACE-N are not present in the inter-face between sACE-C. Instead, the sACE-C interface is formed by a symmetric salt bridge between K1067 and E1060, side chain-backbone interactions, and van der Waals contacts (*Figure 3D*).

## Conformational states of sACE catalytic domains

The comparison of our dimeric sACE cryo-EM structures reveals the conformational dynamics of sACE catalytic domains. The four sACE dimer structures represent different combinations of open, interme-diate, and closed states of sACE catalytic domains (*Figure 2*). By aligning each sACE-N and sACE-C region in our structures to the D2/4 domain, it becomes clear that our structures represent a continu-ous gradient of structural heterogeneity, ranging from the most closed state to the most open state (*Figure 2—figure supplement 2*).

We performed a series of all-atom molecular dynamics (MD) simulations to explore the confor-mational dynamics of sACE. A total of eight apo-sACE simulations were conducted, comprising four simulations with non-glycosylated sACE (>1.8 μs each) and four with sACE where 12 N-linked glyco-sylation sites that contained observable extra density were modeled with complex glycans (>2 μs each; *Figure 4—videos 1–2*). Glycan orientations were randomized prior to each simulation to avoid bias. We did not observe a substantial difference in the global conformational dynamics between our sets of simulations with and without glycans. The domain motions of these simulations are summa-rized in *Supplementary file 3*.

RMSD analysis reveals that the dominant source of conformational change in our simulations is the open-close transition between the D1/3 and D2/4 domains (*Figure 4—figure supplement 1*). The D2/D2 and D4/D4 regions are the most stable, with the bulk of conformational changes localized

to the D1/D3 domain. Within the D1/3 domain, we observed that the D1/3 a motion was not always correlated with D1/3b or D2/D4, meaning that at a given time, D1/3 a could move along with D1/3b, or with D2/D4, or independently. Consistent with our structural analysis, our MD simulations also indicate that sACE-N displays a much wider range of opening geometries than sACE-C.

Most of our MD simulations rapidly closed and remained closed for the duration of the simulation, preventing an effective comparison of the open vs closed state dynamics. However, we were able to garner significant insight into the mechanism of open-close transition by integrating our MD simulations and cryo-EM structural analysis. Initially, we sought to understand the stable and mobile regions of sACE by calculating the Cα displacement for each residue among the four difference cryo-EM structures. As shown in *Figure 4A*, the D2 domain was found to be the most stable region of sACE while the D1 domain was the most mobile, highlighting the impact of the sACE-N open-close transition. sACE-C was, globally, very stable outside of surface-exposed loops and the tip of the D3a subdomain. This analysis predicted the most mobile regions of sACE to be the ends of the D1a subdomain and a loop region with the D1b that closely associates with the tip of the D1a subdomain.

While the structural perturbations predicted by comparing our cryo-EM structures correlate well with the global RMSD analysis of our MD simulations (*Figure 4—figure supplement 1*), we performed further analysis to understand the finer details of sACE conformational dynamics. To best compare the cryo-EM structures with our MD simulations, we compared the calculated alpha carbon displacement of our cryo-EM structures to the root mean square fluctuation (RMSF) values for each residue in our MD simulations (*Figure 4*). We observed consistent, global agreement between our cryo-EM data and MD simulations, both with and without glycans present. This consistency is particularly noticeable in several key mobile regions (denoted 1–4 in *Figure 4*) that demonstrate high degrees of Cα displacement from our cryo-EM data along with correlating higher RMSF during subsets of our MD simulations (*Figure 4A, B and D*). The RMSF variation between stable and mobile regions in our MD simulations is dampened, relative to the cryo-EM data when the entire trajectories were used (*Figure 4A, C and E*). This is due to the open state regions closing relatively early in the simulation and remaining closed for the duration of the simulation. We then performed principal component analysis (PCA; *Figure 4—figure supplement 2*) and cluster analysis (*Figure 4—figure supplement 3*) to compare our MD simulation data with our cryo-EM data. Both methods of analysis indicate that the conformational space sampled by our cryo-EM structures represents a fraction of the conformational space sampled by our MD simulations.

The dynamics observed in our MD simulations of sACE are highly complex, arising from a combination of both interdomain and intradomain motions. To summarize these simulations in simpler terms, individual sACE domains transition from a more open state to a closed state without reopening, while sACE-N/N and sACE-C/C primarily move relative to each other as rigid bodies. As outlined previously, PCA was employed to reduce the complexity of these motions and to identify dominant trends in conformational changes within the dataset. Visualizing the conformational changes captured by the principal component vectors in the MD simulations (*Figure 4—videos 3–4*) revealed that the majority of conformational variability within the MD data is driven by substantial interdomain motion occurring between sACE-N/N and sACE-C/C. Consistent interdomain motion was observed along all principal component vectors, regardless of the presence or absence of intradomain motion. However, no principal components were identified that exclusively described significant intradomain motion without associated interdomain motion. The PCA further revealed that glycosylation significantly enhances the degree of interdomain motion in sACE, as evidenced by the glycosylated simulation (*Figure 4—video 4*) compared to the non-glycosylated simulation (*Figure 4—video 3*). This suggests that glycosylated sACE exhibits greater global dynamics (*Figure 4*, *Figure 4—figure supplement 4*). Notably, principal components 5 and 6 from the glycosylated MD simulations (*Figure 4—video 4*) highlight a striking difference, where sACE-N/N and sACE-C/C appear to move independently and exhibit a decoupled interaction. We postulate that this inter-domain motion may be the reason we observe preferential denaturation of sACE-C during cryo-EM grid preparation. The relatively harsh conditions that the protein is exposed to during the grid-making process may damage specific conformations of particles and subsequently restrict the amount of information that could be obtained solely from cryo-EM analysis. This highlights the importance of integrating multiple approaches to investigate protein dynamics.

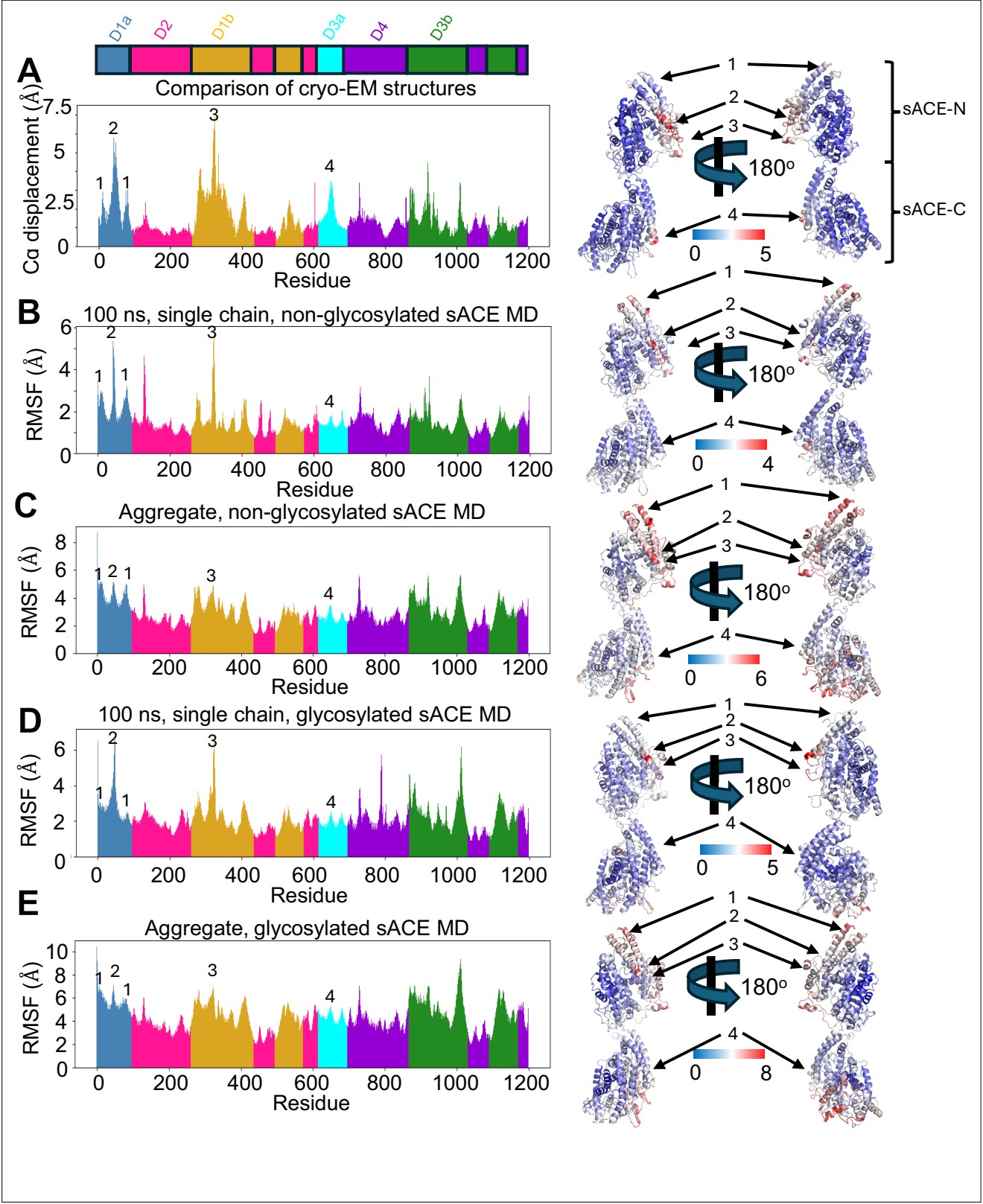

**Figure 4.** Comparison of molecular dynamics simulations with cryo-EM data. (**A**) Alpha carbon displacement values for each residue were calculated by comparing each of our cryo-EM structures against one another in a pairwise manner and averaged. Individual residues are colored by sub-domain as defined in *Figure 1*. Alpha carbon displacement values were also mapped onto the structure of sACE to better visualize the mobile regions. Residues are colored by degree of displacement from blue (no displacement) to red (high displacement, values in Angstrom, as indicated). Numbers denote

*Figure 4 continued on next page*

*Figure 4 continued*

regions of interest: 1, the top of the D1a sub-domain, 2, the bottom of the D1a sub-domain, 3, a flexible loop region within the D1b that interacts with the bottom of the D1a sub-domain near the catalytic cleft, 4, the bottom of the D3A sub-domain. (**B**) Alpha carbon RMSF values for a representative 100 ns subset from our non-glycosylated MD simulations that demonstrates the simulated dynamics correlate well with displacement calculated from the cryo-EM structures. (**C**) Alpha carbon RMSF values for the entirety of our non-glycosylated MD simulations. Mobile regions agree with predictions from the cryo-EM structures, yet the magnitude is dampened due to the protein rapidly transitioning from an open state to a closed state and remaining closed for most of the simulation time. (**D**) Alpha carbon RMSF values for a representative 100 ns subset from our glycosylated MD simulations. (**E**) Alpha carbon RMSF values for the entirety of our glycosylated MD simulations.

The online version of this article includes the following video and figure supplement(s) for figure 4:

**Figure supplement 1.** RMSD analysis of non-glycosylated MD simulations.

**Figure supplement 2.** Principal component analysis.

**Figure supplement 3.** Clustering analysis.

**Figure supplement 4.** RMSF comparison of glycosylated vs non-glycosylated MD simulations.

**Figure supplement 5.** Shielding of glycans in MD simulations.

**Figure 4—video 1.** Non-glycosylated sACE MD simulation.

https://elifesciences.org/articles/106044/figures#fig4video1

**Figure 4—video 2.** Glycosylated sACE MD simulation.

https://elifesciences.org/articles/106044/figures#fig4video2

**Figure 4—video 3.** Non-glycosylated PCA trajectories were generated as linear interpolations representing the conformational changes described along each of the top 10 principal components from our analysis (See *Figure 4—figure supplement 2*).

https://elifesciences.org/articles/106044/figures#fig4video3

**Figure 4—video 4.** Glycosylated PCA trajectories were generated as linear interpolations representing the conformational changes described along each of the top 10 principal components from our analysis (See *Figure 4—figure supplement 2*).

https://elifesciences.org/articles/106044/figures#fig4video4

---

The MD simulations of glycosylated sACE provide additional structural insights into the structure and functions of N-linked glycosylation of sACE. Glycosylation is shown to affect the folding, secretion, and stability of ACE (*Kost et al., 2003*; *O'Neill et al., 2008*; *Sadhukhan and Sen, 1996*). Our MD simulation reveals that the dynamics of 12 N-linked glycans cover a large surface area of sACE, shielding it from environmental insults (*Figure 4—figure supplement 5*). Consistent with observations from our Coulomb potential density maps and previously published data, we observed a substantial and persistent inter-subunit interaction between glycans from N82 (*Figure 3C*). From RMSF analysis, glycosylation was found to increase the overall dynamicism of sACE (*Figure 4—figure supplement 4*). Furthermore, while there are N-linked glycans near the catalytic chamber, particularly those at the tip of D1a and D3a, they were highly dynamic and did not occlude access to the catalytic chamber. This is consistent with the observation that deglycosylation only modestly increases the enzymatic activity of ACE (*Baudin et al., 1997*; *Orth et al., 1998*; *Yu et al., 1997*).

The comparison of sACE-N in its open and closed states reveals the structural basis for the open-to-closed transition of this domain. Our structures show that when the bottom of the D1a subdomain (region of interest 2 in *Figure 4*) moves away from the D2 subdomain, resulting in the open state, the top of the D1a subdomain (region of interest 1 in *Figure 4*) moves closer to the D2 subdomain, as if the helix were balanced upon a fulcrum (*Figure 5A and B*). Detailed structural analysis indicates that the fulcrum is primarily formed by the salt bridge between two highly conserved residues: E66 in the D1a subdomain and R108 in the D2 subdomain. To stabilize the open state, additional salt bridges are formed, with D189 in the D2 subdomain pairing with K73 in all cases and occasionally with R96 in the D1a subdomain (*Figure 5A*). The loss of this salt bridge in the closed state is compensated for by the formation of a network of van der Waals interactions between the D1a and D2 subdomains that stabilize the closed state. The closed state of sACE-C is also stabilized by a network of van der Waals contacts between the D3a and D4 subdomains. The transition between open and closed states is likely influenced by the binding of substrates and reaction products during the ACE catalytic cycle. Although a similar fulcrum mechanism is not evident in the sACE-C domain, the connection between the D3a and D2 subdomains would act as a lever to pivot sACE-C from the closed to the open state, and we speculate that this may form the structural basis for allosteric communication between sACE-N and sACE-C.

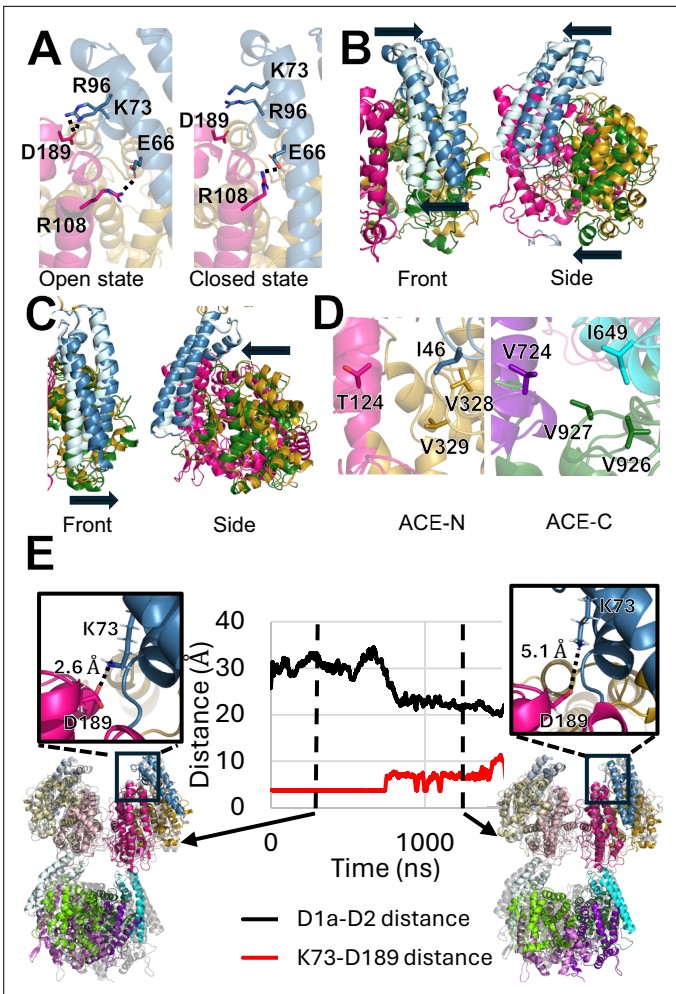

**Figure 5.** Structural mechanism of the sACE open/close transition. (**A**) sACE-N overlay comparing the open (left panel) and closed (right panel) states in detail. The open state is stabilized by interaction between residues in the D1a (blue) and D2 (magenta) regions that, notably, K73-D189. In the closed state, the K73-D189 interaction is broken. (**B**) Overlay of the sACE-N open (dark shades, sub-domains colored as above) and closed (light shades, sub-domains colored as above) states showing the range of motion. The D1a region rotates about a fulcrum region described in (**A**), while the D1b region moves as a rigid body. Front view arrows depict the 'fulcrum' motion of the D1a subdomain, with the top and bottom of the subdomain moving in opposite directions. Side view arrows depict rigid body motion of the D1a and D1b subdomains moving together. (**C**) Overlay of sACE-C closed (light shades, subdomains colored as above) and intermediate (dark shades, subdomains colored as above) states. Unlike sACE-N, the 'top' of the D3a region is constrained by its connection to sACE-N and largely immobile. The primary source of opening is only the motion of the D3a tip. We did not observe any open state structures of sACE-C, suggesting a smaller range of motion relative to sACE-N. Front view arrow depicts motion of D3a subdomain. Only the bottom of the subdomain moves, in contrast to the 'fulcrum' motion observed in the D1a subdomain. Side view depicts the rigid body motion of the D3a and D3b subdomains moving together. (**D**) Comparison of the hydrophobic 'latch' region formed in the closed state between residues of the D1/3 a, D1/3b, and D2/D4 domains. V724 in sACE-C has been replaced by T124 in sACE-N, suggesting that the closed state in sACE-N may be less stabilized than the sACE-C closed state. (**E**) Example all-atom MD simulation tracking the openness of one sACE-N region (black line) and this distance between K73 and D189 (red line). These residues form a salt bridge early in the simulation when sACE-N is open (left inset), but the interaction breaks as sACE-N transitions to the closed state (right inset). Distance measurements for MD simulations were consistently greater than distance values in our static structures and cannot be directly compared to *Figure 2*.

On average, sACE-N is more open than sACE-C, likely due to multiple factors (*Figure 2D*). In sACE-N, the D1a subdomain is at the N-terminal end of sACE, allowing for a greater range of motion (*Figure 5B*), while the fulcrum and stabilization mechanisms can maintain sACE-N in a more widely open state (*Figure 5A*). Conversely, in sACE-C, the motion of the D3a subdomain is constrained by its connection to the D2 subdomain (*Figure 5C*). Additionally, there are sequence differences at the interface between the three subdomains of sACE-N and sACE-C. In the most closed state—the B chain sACE-C domain of the sACE-3.05 structure—residues from all three subdomains, including I649, V724, V926, and V927, form a hydrophobic latch that stabilizes the closed state (*Figure 5D*). The hydrophobic latch in sACE-N is weaker because the residue corresponding to V724 has been replaced with a threonine (T124; *Figure 5D*). These structural insights are supported by our RMSF analysis of our MD simulations and alpha carbon displacement analysis of our cryo-EM structures, which reveal a correlation between mobile regions and the 'fulcrum' mechanism of opening for sACE-N (*Figures 4 and 5A*). Furthermore, our slowest closing MD simulation revealed that the K73-D189 interaction was maintained for the entire time that sACE-N adopted an open conformation, yet this interaction was lost when sACE-N transitioned to the closed state (*Figure 5E*).

We also performed three MD simulations on the non-glycosylated sACE monomer (>1 microsecond each) (*Supplementary file 3*). We observed the expected intra-subdomain open-closed motions in sACE-N and sACE-C that were also observed in the simulations of ACE dimer. We also observed that the motion of sACE-C relative to sACE-N was significantly greater in our monomer simulations compared to our dimer simulations. This suggests that dimerization places significant constraints upon the conformational dynamics of sACE. Previous work has defined the conformational dynamics of sACE from cryo-EM data where both sACE-C domains were not resolved (*Liang et al., 2022*). Given our observations from the monomeric and dimeric MD simulations, we endeavored to understand the conformational heterogeneity present within our cryo-EM data to better understand the impact of dimerization on sACE dynamics.

## Cryo-EM heterogeneity analysis of sACE

Cryo-EM structures are ensemble averages of many aligned particle images. Recently, there have been many approaches developed to understand the particle heterogeneity present within a cryo-EM structure as a proxy for molecular motion (*Tang et al., 2023*). We employed three methods of heterogeneity analysis: cryoDRGN, RECOVAR, and 3D Variability Analysis (3DVA; *Punjani and Fleet, 2023*; *Gilles and Singer, 2024*; *Zhong et al., 2021*). CryoDRGN employs neural networks to resolve structural heterogeneity in a nonlinear, multidimensional latent space, while RECOVAR and 3DVA both employ linear subspace methods. 3DVA finds the linear subspace by an alternating optimization scheme to capture image variations, while RECOVAR utilizes a regularized covariance estimator to identify subspaces that best capture the distribution of states.

We conducted this analysis and observed global agreement between the results of both of our datasets. As we observed substantial particle damage in our Vitrobot-prepared dataset, we thought it unwise to utilize that dataset for in-depth analysis and therefore, we focus on the larger and higher-resolution dataset collected at NCCAT below. For analysis, the entire particle set, prior to 3D classification and consisting of ~170 k particles, was utilized. The particle alignments from the same C1 consensus refinement were used as input for each of the three methods as current methods to investigate particle heterogeneity are highly dependent upon initial pose assignment of individual particles. Globally, all three methods agreed with our ensemble structures, in that the dominant source of structural variability within our particle population could be described by the open-closed transition of individual domains, and that sACE-N displayed more variability than sACE-C.

Initially, we performed 3D variability analysis (3DVA) in cryoSPARC to provide a direct comparison to the previously published analysis of the partial ACE cryo-EM structure (*Punjani and Fleet, 2021*; *Figure 6A*). Trajectories were reconstructed along the top 5 principal components of variance (*Figure 6—video 1*). Surprisingly, the trajectory describing the variance along the first principal component vector suggests a dominant interdomain motion, where the sACE-N/N region bends towards the sACE-C/C region, similar to what we observed for trajectories along the top principal component vectors in analysis of our MD simulations (*Figure 4—videos 3–4*), yet this variance is not reflected in our ensemble structures. The trajectories along the remaining principal component vectors comprise varying combinations of individual domains undergoing the open-close transition.

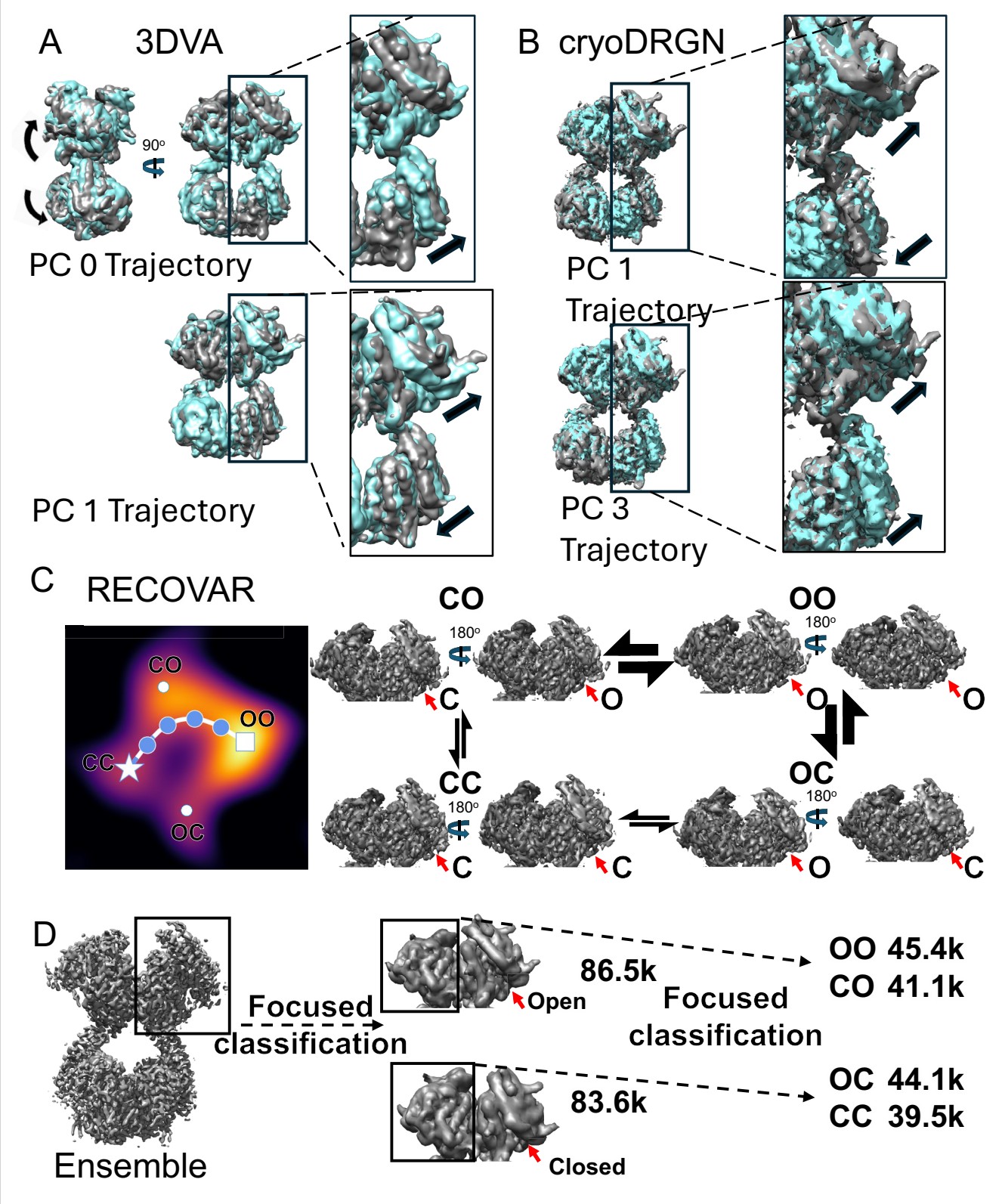

**Figure 6.** Cryo-EM heterogeneity analysis. (**A**) Visualization of the structural changes revealed by cryoSPARC 3DVA trajectories calculated along two principal components (PCs) of structural variance. Starting states are showing in cyan, ending states in gray. PC 0 reveals a large, inter-domain bending motion accompanied by the open/close transition in sACE-C. PC 1 and the remaining PCs are dominated by the open/close transition of individual regions. See **Supplementary file 3** and **Figure 6—video 1** for additional details. Arrows depict generalized motions. (**B**) Visualization of the structural

*Figure 6 continued on next page*

*Figure 6 continued*

changes revealed by the cryoDRGN trajectories calculated along two PCs of structural variance. Starting states are shown in cyan, ending states in gray. PCs are dominated by the open/close transition of individual regions. See **Supplementary file 3** and **Figure 6—video 2** for additional details. Arrows depict generalized motions. (**C**) Analysis in RECOVAR with a focus mask on sACE-N/N reveals that particles adopt roughly four clusters within the latent space (heat map of particle density) corresponding to the open-open (OO, white square), open-closed (OC, white dot), closed-open (CO, white dot), and closed-closed (CC, white star) states of sACE-N/N. A trajectory estimating the path through latent space corresponding to the structural transition from sACE-N/N CC state to the OO state (blue points) suggests that individual sACE regions transition at different rates, as indicated by the size of the transition arrows between states. See **Figure 6—video 3** for trajectory. (**D**) Focused 3D classification was performed in cryoSPARC to explore evidence of coordinated motion between sACE-N regions. 3D classification focusing on one sACE-N region revealed two roughly equal classes of particles: open and closed. Subsequent 3D classification focused on the other sACE-N region again revealed two roughly equal classes, suggesting the lack of coordinated motion between sACE-N regions in the absence of substrate.

The online version of this article includes the following video and figure supplement(s) for figure 6:

**Figure supplement 1.** CryoDRGN clustering of particles.

**Figure supplement 2.** Artificially observed interdomain motion caused by the subtle yet noticeable differences between the two chains within the sACE dimer and the choice of chain pairs between sACE dimer structures.

**Figure 6—video 1.** CryoSPARC 3D variability analysis.

https://elifesciences.org/articles/106044/figures#fig6video1

**Figure 6—video 2.** CryoDRGN heterogeneity analysis.

https://elifesciences.org/articles/106044/figures#fig6video2

**Figure 6—video 3.** RECOVAR trajectory.

https://elifesciences.org/articles/106044/figures#fig6video3

**Figure 6—video 4.** 3DFlex trajectory.

https://elifesciences.org/articles/106044/figures#fig6video4

---

Consistent with our previous observations, the range of presumptive motion in these trajectories is greater in sACE-N than sACE-C. Previous 3DVA performed on the sACE monomer structure revealed 4 types of potential motion, termed bending, pivoting, breathing, and jumping (**Lubbe et al., 2022**). The breathing motion matches the open-close transition that dominates most of our principal component vectors, while the bending motion appears to correlate with the variance described by our top principal component vector. We did not observe any potential motions reminiscent of the jumping or pivoting motions in our 3DVA. Furthermore, the magnitude of the bending motion in our 3DVA is substantially less than previously reported for the partial dimeric sACE structure, which likely suffered from substantial denaturation (**Lubbe et al., 2022**). Together, this highlights a significant difference in the conformational dynamics between our sACE dimer and previously reported cryo-EM structures. While there is disagreement between the 3DVA results for the two datasets, it should be noted that our MD simulations display a greater range of motion than either cryo-EM dataset offers, and the varieties of motion described previously are clearly observed within the PCA trajectories, especially when sACE is glycosylated (**Figure 4—video 4**). As mentioned above, this type of cryo-EM analysis is sensitive to artifacts introduced by particle damage and denaturation, underscoring the importance of preserving protein integrity via sample preparation and the use of Chameleon to reduce the exposure to air-water interface. This also suggests dimerization functions as a key constraint to the conformational dynamics of sACE.

We then moved on to analyze our data using cryoDRGN, which revealed that the data formed a single cluster across multiple dimensions (**Figure 6**, **Figure 6—figure supplement 1**). Trajectories generated along five principal component vectors of structural variance were dominated by the open-close transition, primarily in sACE-N (**Figure 6—video 2**). We observed little variance in sACE-C and, unlike 3DVA, cryoDRGN did not reveal any substantial interdomain motions. Faced with this discrepancy between the 3DVA and cryoDRGN results, we employed a third method, RECOVAR, to further analyze our data. Similar to cryoDRGN, RECOVAR revealed that our data formed a singular cluster across multiple dimensions of analysis. Volumes representative of k-means cluster reconstructions were similarly consistent with our cryoDRGN results. RECOVAR indicated that the primary source of structural variance could be explained by the open-close transition, primarily of the sACE-N domains. With this in mind, we repeated the RECOVAR analysis with a focus mask around the sACE-N/N region. We then estimated the conformational density by deconvoluting the multidimensional latent space

with a two-dimensional principal component analysis. The resulting density plot revealed four clusters that corresponded to different conformations of the sACE-N/N region (*Figure 6C*). The most populous cluster corresponded to a state where both sACE-N subunits adopted an open conformation. The second largest cluster, and the smallest cluster, corresponded to a state where one sACE-N was open and the other was closed. The discrepancy in population size between these clusters is likely due to bias in the initial particle orientation, rather than a subunit-specific preference for the open state. As the O/C state and the C/O state are 180 degree rotations of each other, particle assignment to either cluster is likely influenced by the initial particle orientation of the complete dimer, and we currently lack the data to discern any allosteric implication to the orientation assignment. The final cluster corresponded to a state where both sACE-N subunits adopted the closed conformation. We then used RECOVAR to generate a trajectory through the latent space to predict a potential transition pathway from the closed-closed state to the open-open state (*Figure 6—video 3*). This trajectory revealed that one subunit would open, followed by the second, and raised the question of potential allostery or coordinated motion between the adjacent sACE-N domains.

To investigate coordination between the sACE-N domains, we performed 3D classification in cryoSPARC with a focus mask on a single sACE-N domain and requested two classes (*Figure 6D*). The particles were split roughly evenly between the two classes, one resulting in an open state and the other being closed. We then used these classes for another round of 3D classification with a focus mask on the opposite sACE-N domain and again asked for two classes. Again, these particles were split roughly evenly among the classes, leaving us with four similarly sized classes adopting the same conformations as the clusters revealed by RECOVAR.

## Discussion

Our cryo-EM and MD simulations analysis of the extracellular regions of dimeric ACE (sACE) offers insights into the stability of sACE catalytic domains. We show that the sACE dimer interface consists of three major components, protein-protein interfaces between sACE-N, glycan-glycan interaction between sACE-N, and protein-protein interface between sACE-C. The interface between sACE-N, in general, is 50% larger and has higher numbers of hydrogen bonds and salt bridges than that between sACE-C. The necessity for the stronger interaction between sACE-N is likely due to the fact that sACE-N is distal to the transmembrane helix so that the additional reinforcement is required. The weaker interaction and lack of contribution from glycans in the interface between sACE-C could also contribute to the observed preferential denaturation of sACE-C despite the high structural similarity between sACE-N and sACE-C. The denaturation is likely due to the exposure to air-water interface during vitrification, a common issue in single particle cryo-EM analysis (*Glaeser, 2018*; *Noble et al., 2018*). We have partially overcome the denaturation issue of sACE-C by optimizing the buffer pH to increase the melting temperature and using the faster plunging time during vitrification afforded by Chameleon to reduce the exposure to air-water interface (*Figure 1*). However, we did not completely prevent the preferential denature of sACE-C during vitrification.

ACE is part of a family of proteases capable of effectively degrading Aβ. This family includes M16 metalloproteases like IDE and PreP, and M13 metalloproteases such as neprilysin and endothelin converting enzyme. Integrative structural analysis has demonstrated how IDE employs conformational dynamics, charge, and secondary structure complementarity to selectively capture peptide substrates into its sizable catalytic chamber, subsequently unfolding and degrading amyloidogenic peptides such as insulin and Aβ, without degrading structurally similar yet non-amyloidogenic peptides (*Malito et al., 2008*; *Zhang et al., 2018*; *Tang, 2016*). The other mentioned enzymes also belong to this group of chamber-containing proteases, known as cryptidases (*Liang et al., 2022*; *Malito et al., 2008*; *Liang et al., 2021*). Similar to cryptidases, structural analysis reveals that ACE must undergo an open-closed transition to capture, engulf, and degrade its oligopeptide substrates, such as angiotensin 1 and bradykinin.

Our apo-sACE structures and MD simulations identify a fulcrum mechanism that governs the open-closed transition of sACE-N, while the interplay between sACE-N and the D3a domain likely plays a crucial role in the corresponding transition in sACE-C. Analogous to how the binding of MB60, a PreP inhibitor, to the hydrophobic pocket triggers the open to closed transition, factors such as binding of substrates and ACE inhibitors may alter the balance, leading to the regulation of ACE activity (*Liang et al., 2022*). We also observe that the catalytic pocket of each sACE catalytic domain is not large

enough to engulf the entire Aβ. Additionally, the distances between the domains (sACE-N to sACE-C, sACE-N to sACE-N, or sACE-C to sACE-C) are too large for these domains to bind Aβ simultaneously (*Chen et al., 2017*). Therefore, ACE likely employs a different mechanism from IDE and PreP to selectively bind and degrade Aβ. Future integrative structural studies await to reveal how ACE selectively recognizes and degrades a diverse array of substrates.

Globally, we observed that sACE-N is both more open and more dynamic than sACE-C in the absence of substrate. We speculate that the increased dynamicism of sACE-N may be due to a weakening of the hydrophobic latch region relative to sACE-C, which renders the closed state more thermodynamically favorable in sACE-C, along with increased stabilization of the sACE-N open state through the fulcrum mechanism described above. These factors result in sACE-N demonstrating a preference for the open state, while sACE-C appears to prefer the closed state, in the absence of substrate. The preference of sACE-C towards the closed state may explain why it displays a greater catalytic activity compared to sACE-N for substrates such as angiotensin-I (*Zhang et al., 2018*). The difference in dynamics between sACE-N and sACE-C also offers new pathways forward in the development of domain-specific inhibitors to better modulate ACE function. The fulcrum-associated region centered on the K73-D189 interaction stands out as a promising new druggable target to modulate the openness of sACE-N independent of sACE-C. Additionally, given the range in size of ACE substrates, that is amyloid β (42 residues) vs bradykinin/angiotensin-I (9/10 residues), it is likely that the openness of individual ACE domains functions as a key mediating step in substrate recognition and altering the open-to-close dynamics of ACE domains may prove an effective strategy to modulate ACE activity towards specific subsets of substrates. These insights come with the caveat that they are derived from the apo-state dynamics of sACE, and it remains unknown how sACE dynamics are influenced by the presence of substrate. Studies in the presence of substrate will be especially vital in exploring potential mechanisms of ACE allostery.

The interplay between ACE-N and ACE-C within the ACE dimer has complicated consequences for ACE function, showing negative cooperativity for enzymatic activities while exhibiting positive synergy in Aβ cleavage by ACE-N and activity at the bradykinin receptor (*Zou et al., 2009*; *Binevski et al., 2003*; *Rice et al., 2004*; *Skirgello et al., 2005*; *Erdös et al., 2010*). Understanding the conformational dynamics in both inter- and intradomain motions of the extracellular regions of the ACE dimer holds promise for designing ACE modulators to improve and broaden the therapeutic use of ACE modulation. Our cryo-EM heterogeneity analysis of sACE alone has revealed coordinated yet complicated pattern of open-closed transitions across four catalytic domains using cryoDRGN and cryoSPARC 3DVA (*Figure 6A and B*, *Figure 6—videos 1–2*). RECOVAR analysis further illustrates the distribution of open-closed states within the sACE-N dimer and the potential pathways for such transitions (*Figure 6C*). Additionally, focused classification using cryoSPARC suggests no bias in the conformational state of sACE-N within a dimer, irrespective of the state of its counterpart (*Figure 6D*). Altogether, these findings establish a framework for understanding how the complex open-closed transitions within the four catalytic domains of the sACE dimer contribute to its allosteric behavior (*Zou et al., 2009*; *Binevski et al., 2003*; *Rice et al., 2004*; *Skirgello et al., 2005*; *Erdös et al., 2010*).

3DVA, cryoDRGN, and RECOVAR all appeared to display varying degrees of coordinated motion between sACE domains. However, this type of analysis lacks an explicit time component, so it is not possible to draw concrete conclusions regarding coordinated motion. Likewise, while MD simulations contain an explicit time factor, most of our MD simulations closed quickly and did not reopen. While we observed an apparent association between intradomain and interdomain motion in our PCA (*Figure 4—videos 3–4*), the analysis removes time dependency. Furthermore, the aforementioned allostery of ACE was observed under the condition where substrate is present (*Zou et al., 2009*; *Binevski et al., 2003*; *Skirgello et al., 2005*). Complicating matters further, this conformational heterogeneity appears to stem partially from subtle yet significant conformational differences between the two chains within an ACE dimer and the inability of cryoSPARC to accurately align these conformationally distinct chains, likely due to particle pseudo-symmetry (*Figure 6—figure supplement 2*). Furthermore, CryoDRGN and RECOVAR revealed smaller magnitudes of potential interdomain motions than cryoSPARC 3DVA, and they were often accompanied by various combinations of individual domains opening and/or closing. This suggests that the apparent discrepancy in the magnitude of inferred interdomain motion could be a result of the different theoretical approaches of dimensionality reduction and trajectory generation employed by the respective methods of analysis

and underscores the importance of incorporating multiple methods of analysis in the absence of a singular 'gold-standard' of heterogeneity analysis. Conversely, it is possible that the interpretation of any potential interdomain motion is complicated by the preferential denaturation of sACE-C. PCA of our MD simulations revealed the dominant source of motion to be alterations in the position of sACE-C/C relative to sACE-N/N. We speculate that many of these relative orientations cannot tolerate the rigors of the cryo-EM grid-making process and may be the source of the observed preferential denaturation of sACE-C. Future studies will need to focus on cryo-EM analysis of ACE in the presence of substrates, ideally using data in which partial denaturation is substantially minimized.

Single particle cryo-EM holds promise for revealing the conformational dynamics of biological macromolecules from embedded heterogeneity. Consequently, cryo-EM heterogeneity analysis is an active and rapidly evolving field providing insights into the conformational dynamics and allostery of macromolecules (*Tang et al., 2023*). However, many challenges exist in using cryo-EM hetero-geneity analysis due, in part, to the low signal-to-noise ratio in observed micrographs. Often, these analyses address questions difficult or impossible to tackle experimentally, leaving no ground truth for validation. We encounter such challenges in our heterogeneity analysis as well. Consistent with our single particle cryo-EM structural analysis, we observed the expected intradomain open-closed transition using cryoDRGN, cryoSPARC 3DVA, RECOVAR, and all-atom MD simulation. Surprisingly, we did not observe such motion using cryoSPARC 3DFlex, a neural network-based method analyzing our cryo-EM data of sACE (*Punjani and Fleet, 2023*). Central to the working of cryoSPARC 3Dflex is the generation of a tetrahedral mesh used to calculate deformations within the particle population. Proper generation of the mesh is critical for obtaining useful results and must often be determined empirically. Despite several attempts, we were unable to obtain results from 3Dflex comparable to what we observed with our other methods. Even using the results from our 3DVA as prior input to 3Dflex, the largest conformational change we observed was a slight wiggling at the bottom of the D3a subdomain (*Figure 6—video 4*). The authors of 3Dflex note that 3Dflex struggles to model intri-cate motions, and the implementation of custom tetrahedral meshes currently requires a non-cyclical fusion strategy between mesh segments. Given these limitations, and the complexity of sACE confor-mational dynamics, it appears that sACE, as a system, is not well-suited to analysis via 3Dflex in its current implementation. Similarly, MDSPACE holds tremendous promise as a method for investigating conformational dynamics from cryo-EM data (*Vuillemot et al., 2023*). MDSPACE integrates cryo-EM particle data with short MD simulations to fit atomic models into each particle image through an iter-ative process which extracts dynamic information. However, the lack of GPU-enabled processing for MDSPACE requires either a dedicated computational setup that diverges from most other cryo-EM software, or access to a CPU-based supercomputer, which severely limits the accessibility of such software. Despite these challenges, both 3Dflex and MDSPACE use promising approaches to study protein conformational dynamics. We look forward to exploring effective methods to incorporate these strategies into our future research.

Advances in single-particle cryo-EM analysis have enabled detailed structural examination of the sACE dimer. Beyond its implications for future drug discovery efforts aimed at enhancing ACE-based therapies, sACE can serve as a valuable model of a multi-domain glycoprotein for studying methods to preserve sample integrity, particularly due to the high propensity of sACE-C to denature during vitrification. Furthermore, considering the aforementioned challenges and caveats, we propose that, due to its size, biomedical importance, and the complexity of its motions, sACE also represents an excellent model system for evaluating and improving cryo-EM heterogeneity analysis.

## Materials and methods
### Expression of human sACE

Human 293T and 293 F cells were obtained from ATCC (CRL-3216) and Thermo Fisher (R79007), respectively. Cell lines were subjected to re-authentication on a regular basis to ensure no cross-contamination. All cells are routinely screened for the presence of mycoplasma using the ATCC Universal Mycoplasma Detection Kit (Catalogue # 30–1012 K). These cells were maintained in DMEM +10% fetal calf serum (FCS). The lentiviral plasmid for the expression of the extracellular domain of human ACE (sACE, aa 1–1235) that contains a 6-histidine tag at its C-terminus was made by Vector-builder and co-transfected with helper plasmids to human 293T cells for lentiviral production. Human

293 F cells were infected with the resulting virus and selected for stable sACE expression. 293 F that stably expressed sACE grew poorly in Free-style 293 media (ThermoFisher 12338018). Thus, for ACE production, 293 F cells that stably expressed sACE were first grown in DMEM +10% FCS at 37 °C and gradually shifted into Free-style 293 media by reducing FCS by first in Free-style 293 media +2% FCS and then to media without FCS. These cells were then shifted to and maintained in 30 °C for 3–5 days for ACE production to reduce proteolysis.

## Purification of soluble human ACE

Culture supernatant was centrifuged at 3000x $g$ for 5 min, diluted 10-fold with 25 mM Tris-HCl pH 8, filtered, and run over a Source Q column pre-equilibrated with 25 mM Tris-HCl pH 8. The column was washed to baseline with Tris-HCl pH 8 and bound protein was eluted with a linear gradient of NaCl to 1 M. sACE containing fractions were pooled and EDTA was added to 10 mM to strip the catalytic zinc ion. The sample was concentrated and run over a Superdex S200 SEC column pre-equilibrated with the desired cryo-EM buffer.

## Differential scanning fluorometry (DSF)

To optimize the grid making conditions, differential scanning fluorimetry was applied to screen 96 buffers conditions. DSF was performed with 3 mg/ml sACE and 10 X Sypro Orange in 20 µl buffers in a Thermo Fisher Step ONE RT-PCR thermocycler. Using the melting temperature ($T_M$) and slope as selective criteria, 25 mM citrate pH 5.5, 25 mM sodium phosphate pH 5.5, 150 mM NaCl, 10 mM EDTA was identified as the optimal condition for the highest $T_M$ and optimal slope, which was used for all subsequent cryo-EM experiments.

## Cryo-EM data collection and processing

Peak fractions were collected off the SEC column and used to immediately prepare grids, resulting in a protein concentration of ~4.5 mg/ml. Quantifoil R1.2/1.3 Cu200 grids were glow-discharged in air and grids were prepared by Vitrobot mark 3. 3.5 µl of sample was applied to the grid in the chamber with 100% humidity, 25 °C. Excess sample was blotted for 1 s, blot force 2, and grids were plunge frozen in liquid ethane. All images were acquired using a Titan Krios microscope (FEI) operated at 300 KeV with a Gatan K3 direct electron detector. A dataset of ~3600 micrographs was collected at The University of Chicago Advanced Electron Microscopy facility. The dataset was processed using cryoSPARC version 4.5.1 and 4.6.0 (*Figure 1—figure supplement 2*; *Punjani et al., 2017*). Briefly, a four-domain sACE dimer model was generated using sACE-N dimer (PDB 7Q4D) and sACE monomer (PDB 7Q3Y) and adjusting the distance between ACE-C domains to avoid steric clashes as a template for particle picking (*Lubbe et al., 2022*). After template picking, 1.1 million particles were extracted with a box size of 360 pixels. Several rounds of 2D classification were used to remove junk particles. ~577 K particles were processed in successive rounds of *ab initio* reconstruction and heterogeneous refinement. sACE grids were also prepared using Chameleon and collected a dataset of ~19,000 micrographs at the National Center for Cryo-EM Access and Training. For those, Quantifoil Active R 1.6/0.9 Cu/Rh 300 was coated with gold using a Safematic CCU-010 Compact Coating Unit. Briefly, grids were placed nanowire side down on a glass slide, gold disks were sputtered with a 35 nm target thickness at $1.5 \times 10^{-2}$ Torr, and a process current of 30 mA. Grids were glow discharged with air at 12 mA for 80 s using the internal Chameleon glow discharger. The grids were plunged at 160–180 ms. All images were acquired using a Titan Krios microscope (FEI) operated at 300 KeV with a Gatan K3 direct electron detector (Gatan). Images were automatically acquired using Leginon (*Suloway et al., 2005*) using collection parameters as shown in *Supplementary file 2*. This dataset was processed in cryoSPARC version 4.5.1 and 4.6.0 (*Figure 1—figure supplement 4*). Briefly, templates were generated from our full-length ACE structure described above using vitrobot-prepared grids. Micrographs with a CTF fit resolution worse than 4 Å were removed from the dataset, leaving ~12,000 micrographs for template picking, which yielded 10.7 million particles. The particle set was trimmed to ~2.8 million based on quality metrics, particles were extracted with a box size of 256 pixels and run through 2 rounds of 2D classification, reducing the particle number to ~660,000. 2D classes were dominated by 4 domain views and yielded a much greater diversity of views, indicating that Chameleon preparation produced a dramatic improvement in particle quality. Initial 3D homogeneous refinement reached 3.6 Å, matching the previous final structure. Four maps from 3D classes that contain all four catalytic

domains of ACE, one from particles derived from grids prepared by using vitrobot and three from those prepared by using Chameleon, were used for structure building using Chimera and Coot, and the structure was refined using Phenix real space refinement. Structural analysis was performed using PYMOL and Chimera.

### Heterogeneity analysis

All heterogeneity analysis was performed on the full final particle set, encompassing all of the NCCAT structures (*Supplementary file 4*). Initially, cryoSPARC 3DVA was performed on both the NCCAT- and UChicago-collected datasets, and results were found to be consistent, so the NCCAT-collected dataset was used for further analysis, given its higher particle number and quality. Input particle poses for each method of heterogeneity analysis were derived from the same C1 refinement job in cryo-SPARC, which reached 2.88 Å. 3DVA was used to solve the top 5 modes of variance with a filter resolution of 6 Å. Particles were downsampled to a box size of 128 pixels in cryoDRGN 3.3.1 to speed up processing. The same downsampled particle set was used for both cryoDRGN and RECOVAR. CryoDRGN was run with default parameters and the analysis was performed on the top 5 principal components for direct comparison to cryoSPARC 3DVA. RECOVAR pipeline.py was initially run on the full sACE structure. Reconstructions representative of k-means cluster centers revealed the dominant source of structural variability to be the open-close transition in sACE-N. As a result, RECOVAR was run again with a focus mask on sACE-N/N and a zdim = 4. Conformational density was estimated with a two-dimensional PCA, and an index value of 3 was chosen for further analysis. The points in latent space representing the top 12 stable conformational states were estimated from the top 20% of particles and representative volumes were generated. The generated states populated four clusters, and the 12 states were simplified to 4 distinct conformations where sACE-N/N adopted the open-open (OO), open-closed (OC), closed-open (CO), and closed-closed (CC) states (*Figure 6C*). We then generated a trajectory through the latent space using the OO and CC representative volumes as endpoints (*Figure 6—video 3*).

### All-atom MD simulations and analysis

The sACE-3.65 structure was used to initiate all MD simulations. For the glycosylated sACE simulations, N-linked glycans were built onto all 12 known N-linked glycosylation sites (residues 9, 25, 45, 82, 117, 289, 416, 480, 648, 666, 685, 731) with the complex glycan BGLC-BGLC-BMAN-AMAN-BGLC-ANE5-AMAN-BGLC-BGAL-ANE5-AFUC using CHARMM-GUI (*Lee et al., 2016*). Simulations were prepared in QwikMD or CHARMM-GUI and run at pH 7, 150 mM NaCl, 310 K, 1 atm with periodic boundary conditions in NAMD3.0 using the University of Chicago Beagle3 partition (*Phillips et al., 2020*). Explicit solvent was described with the TIP3P model (*Jorgensen et al., 1983*). Data was analyzed in VMD 1.9.4 (*Humphrey et al., 1996*) using a combination of built-in plugins and custom Python scripts. Analysis scripts are available as source files for the analysis of PCA (*Supplementary file 5*), cluster analysis (*Supplementary file 6*), C-alpha displacement (*Supplementary file 7*).

## Acknowledgements

We are grateful to Mahira Aragon for grid preparation using Chameleon, Kasahun Neselu for data collection, Marc Auréle Gilles for assistance with RECOVAR analysis, and Sergei Danilov for helpful suggestions. This work was supported by the NIH grant GM121964 to W-J Tang. Some of this work was performed at the National Center for Cryo-EM Access and Training (NCCAT) and the Simons Electron Microscopy Center located at the New York Structural Biology Center, supported by the NIH Common Fund Transformative High Resolution Cryo-Electron Microscopy program (U24 GM129539) and by grants from the Simons Foundation (SF349247) and NY State Assembly.

# Additional information

## Funding

| Funder | Grant reference number | Author |
|--------|------------------------|--------|
| National Institute of General Medical Sciences | R01 GM121964 | Wei-Jen Tang |

The funders had no role in study design, data collection and interpretation, or the decision to submit the work for publication.

## Author contributions

Jordan M Mancl, Conceptualization, Data curation, Formal analysis, Investigation, Methodology, Writing - original draft, Project administration, Writing - review and editing; Xiaoyang Wu, Data curation; Minglei Zhao, Data curation, Formal analysis; Wei-Jen Tang, Conceptualization, Data curation, Formal analysis, Supervision, Funding acquisition, Investigation, Writing - original draft, Project administration, Writing - review and editing

## Author ORCIDs

Jordan M Mancl  https://orcid.org/0000-0003-3368-6275
Xiaoyang Wu  https://orcid.org/0000-0001-6378-3207
Minglei Zhao  https://orcid.org/0000-0001-5832-6060
Wei-Jen Tang  https://orcid.org/0000-0002-8267-8995

Reviewer #1 (Public review): https://doi.org/10.7554/eLife.106044.4.sa1
Reviewer #2 (Public review): https://doi.org/10.7554/eLife.106044.4.sa2
Reviewer #3 (Public review): https://doi.org/10.7554/eLife.106044.4.sa3
Author response https://doi.org/10.7554/eLife.106044.4.sa4

# Additional files

## Supplementary files

Supplementary file 1. List of buffers used in DSF optimization of grid-making conditions.

Supplementary file 2. Cryo-EM processing statistics for both datasets.

Supplementary file 3. Summary of MD simulations.

Supplementary file 4. Summary of heterogeneity analysis.

Supplementary file 5. Python script used to perform Principal Component Analysis.

Supplementary file 6. Python script used to perform cluster analysis.

Supplementary file 7. Python script used to perform C-alpha displacement analysis.

MDAR checklist

## Data availability

Unprocessed micrographs, particle stacks, and associated pose/CTF information have been deposited to EMPIAR under the accession numbers: EMPIAR-12181 (Vitrobot-prepared grids) and EMPIAR-12484 (Chameleon-prepared grids) EM structures and maps have been deposited to the EMDB and PDB, respectively, under the following accession numbers: sACE-2.99: 9D5S and EMD-46581 sACE-3.05: 9D5M and EMD-46579 sACE-3.15: 9D55 and EMD-46574 sACE-3.65: 9CLX and EMD-45733.

The following datasets were generated:

| Author(s) | Year | Dataset title | Dataset URL | Database and Identifier |
|---|---|---|---|---|
| Mancl JM, Tang WJ | 2025 | Apo ACE full dimer 3 prepared by chameleon | https://www.emdataresource.org/EMD-46581 | EMDataResource, EMD-46581 |
| Mancl JM, Tang WJ | 2025 | Single particle CryoEM analysis of full length ACE | https://www.ebi.ac.uk/empiar/EMPIAR-12181/ | EMPIAR, EMPIAR-12181 |
| Mancl JM, Tang WJ | 2025 | Apo ACE full dimer 1 prepared by chameleon | https://www.emdataresource.org/EMD-46579 | EMDataResource, EMD-46579 |
| Mancl JM, Tang WJ | 2025 | Apo ACE full dimer 2 prepared by chameleon | https://www.emdataresource.org/EMD-46574 | EMDataResource, EMD-46574 |
| Mancl JM, Tang WJ | 2025 | Angiotensin I converting enzyme full-length dimer | https://www.emdataresource.org/EMD-45733 | EMDataResource, EMD-45733 |

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
