## [Editor Report · eLife Assessment]

This study shows, for the first time, the structure and snapshots of the dynamics of the full-length soluble Angiotensin-I converting enzyme dimer. The combination of structural and computational analyses provides **compelling** evidence that reveals the conformational dynamics of the complex and key regions mediating the conformational change. This **fundamental** work illustrates how conformational heterogeneity can be used to gain insights into protein function.

---

## [Referee Report · Reviewer #1 (Public review)]

Summary:

The authors report four cryoEM structures (2.99 to 3.65 Å resolution) of the 180 kDa, full-length, glycosylated, soluble Angiotensin-I converting enzyme (sACE) dimer, with two homologous catalytic domains at the N- and C-terminal ends (ACE-N and ACE-C). ACE is a protease capable of effectively degrading Aβ. The four structures are C2 pseudo-symmetric homodimers and provide insight into sACE dimerization. These structures were obtained using discrete classification in cryoSPARC and show different combinations of open, intermediate, and closed states of the catalytic domains, resulting in varying degrees of solvent accessibility to the active sites.

To deepen the understanding of the gradient of heterogeneity (from closed to open states) observed with discrete classification, the authors performed all-atom MD simulations and continuous conformational analysis of cryo-EM data using cryoSPARC 3DVA, cryoDRGN, and RECOVAR. cryoDRGN and cryoSPARC 3DVA revealed coordinated open-closed transitions across four catalytic domains, whereas RECOVAR revealed independent motion of two ACE-N domains, also observed with cryoSPARC focused classification. The authors suggest that the discrepancy in the results of the different methods for continuous conformational analysis in cryo-EM could results from different approaches used for dimensionality reduction and trajectory generation in these methods.

Strengths:

This is an important study that shows, for the first time, the structure and the snapshots of the dynamics of the full-length sACE dimer. Moreover, the study highlights the importance of combining insights from different cryo-EM methods that address questions difficult or impossible to tackle experimentally, while lacking ground truth for validation.

Weaknesses (from the last round of review):

The open, closed, and intermediate states of ACE-N and ACE-C in the four cryo-EM structures from discrete classification were designated quantitatively (based on measured atomic distances on the models fitted into cryo-EM maps). Unfortunately, atomic models were not fitted into cryo-EM maps obtained with cryoSPARC 3DVA, cryoDRGN, and RECOVAR, and the open/closed states in these cases were designated based on a qualitative analysis.

---

## [Referee Report · Reviewer #2 (Public review)]

The manuscript presents a valuable contribution to the field of ACE structural biology and dynamics by providing the first complete full-length dimeric ACE structure in four distinct states. The study integrates cryo-EM and molecular dynamics simulations to offer important insights into ACE dynamics. The depth of analysis is commendable, and the combination of structural and computational approaches enhances our understanding of the protein's conformational landscape.

---

## [Referee Report · Reviewer #3 (Public review)]

Summary:

Mancl et al. report four Cryo-EM structures of glycosylated and soluble Angiotensin-I converting enzyme (sACE) dimer. This moves forward the structural understanding of ACE, as previous analysis yielded partially denatured or individual ACE domains. By performing a heterogeneity analysis, the authors identify three structural conformations (open, intermediate open, and closed) that define the openness of the catalytic chamber and structural features governing the dimerization interface. They show that the dimer interface of soluble ACE consists of an N-terminal glycan and protein-protein interaction regions, as well as C-terminal protein-protein interactions. Further heterogeneity mining and all-atom molecular dynamic simulations show structural rearrangements that lead to the opening and closing of the catalytic pocket, which could explain how ACE binds its substrate. These studies could contribute to future drug design targeting the active site or dimerization interface of ACE.

Strengths:

The authors make significant efforts to address ACE denaturation on cryo-EM grids, testing various buffers and grid preparation techniques. These strategies successfully reduce denaturation and greatly enhance the quality of the structural analysis. The integration of cryoDRGN, 3DVA, RECOVAR, and all-atom simulations for heterogeneity analysis proves to be a powerful approach, further strengthening the overall experimental methodology.

Weaknesses:

No weaknesses noted.

---

## [Author Response]

The following is the authors’ response to the previous reviews

We would like to thank you and your chosen reviewers for the diligent work and insightful comments. Following the latest round of feedback, we have made the following changes to the manuscript:

(1) We have added details regarding the specific versions of Cryosparc and cryoDRGN used in our analysis.

(2) We have addressed Reviewer 2’s comment concerning the negative RMSF values in Figure S12. The negative values occur because this display shows the difference in RMSF values from the MD simulations of glycosylated versus non-glycosylated ACE. To avoid similar confusion, we have split Figure S12 into three panels. Panels A and B show the RMSF values for each residue in the glycosylated and non-glycosylated sACE MD simulations, respectively, and all values here are positive. Panel C (the original Figure S12) now includes expanded labeling to clarify that it depicts the difference in RMSF values between the presence and absence of glycans. In this panel, a negative value indicates that the residues exhibit higher RMSF in simulations where glycans are present. The figure legend has been revised to accurately describe the updated figure.